# AI adoption in E-commerce enterprises: Insights into current practices and future directions from an interview study

Tong Zhu[1]*, Mohd Zaidi Abd Rozan[2]

1 International Business School, Anhui International Studies University & Faculty of Management, Universiti Teknologi Malaysia, Johor Bahru, Malaysia, 2 Department of Applied Computing & Artificial Intelligence, Faculty of Computing, Universiti Teknologi Malaysia, Johor Bahru, Malaysia

* zhutong@graduate.utm.my

## Abstract

With the accelerated integration of artificial intelligence (AI) technology in digital commerce, e-commerce businesses are showing a diversified trend in its application across processes such as marketing, operations, and customer services. This article, based on the TOE (Technology-Organization-Environment) model and the TAM (Technology Acceptance Model) framework, employs a multi-case qualitative interview approach to conduct semi-structured interviews with different types of e-commerce enterprises in Anhui Province, China. It explores their practical implementation paths, feedback on effects, and future plans regarding AI technology adoption. The research findings are as follows: First, AI applications have covered the entire chain of e-commerce operations, including content generation, advertising placement, data analysis, customer service management, and logistics scheduling; however, small and micro enterprises still face significant limitations in technical depth and customization capabilities. Second, while the effects of AI applications are emerging, most processes continue to rely heavily on human collaboration and supervision, resulting in a human-machine collaboration model of "AI pre-processing + human fine-tuning." Third, the organizational capabilities of enterprises and the AI literacy of employees are key to adoption, while the external policy environment has yet to provide effective guidance. Fourth, enterprises exhibit a stratified integration pattern in AI deployment, ranging from 'platform-bound' to 'tool-augmented' to 'self-developed' models, reflecting significant dynamic variations across different organizational contexts. This study also provides a new theoretical perspective and empirical evidence for understanding the technology adoption logic and digital transformation practices in the AI-driven e-commerce industry.

**Data availability statement:** All data underlying the findings have been deposited in a stable public repository (Zenodo) and are freely accessible at: (DOI: 10.5281/zenodo.18298170). The dataset is fully anonymized and does not contain personal identifiers or sensitive information.

**Funding:** This research is supported by the project "Application Research of Artificial Intelligence Technology in e-Commerce Enterprises in Anhui Province", with Fund Number 2024AH052502. The funder had no role in the study design, data collection and analysis, decision to publish, or preparation of the manuscript.

**Competing interests:** The authors have declared that no competing interests exist.

## Introduction

In recent years, the rapid advancement of digital technology has led to the widespread integration of artificial intelligence (AI) across various facets of e-commerce. AI has become a crucial tool for companies seeking to optimize operational efficiency, enhance user experience, and enable intelligent decision-making [1]. Whether in product content generation, advertising placement, or key processes such as customer service, data analysis, and intelligent logistics, AI technology has demonstrated significant empowering potential for e-commerce enterprises. These enterprises encompass diverse forms, including platform-based merchants, self-operated stores, cross-border e-commerce, and live-streaming e-commerce, all conducting commercial transactions on digital platforms through internet technology [2].

However, existing studies reveal that enterprises differ substantially in how AI technologies are applied, integrated, and embedded into e-commerce operations, particularly across firms of different sizes and digital capacities [3]. Although prior research has documented the performance benefits of AI adoption and examined user-level acceptance mechanisms, much of the literature remains either technology-centric, emphasizing efficiency gains, or behavior-centric, focusing on individual adoption intentions [4,5]. Relatively less attention has been paid to how e-commerce enterprises, in practice, deploy AI across different platforms and business processes, evaluate its application outcomes, and adjust usage strategies under concrete platform, organizational, and environmental conditions. As a result, existing research provides limited insight into the diversity of AI application patterns observed across different types of e-commerce enterprises, particularly regarding how operational processes, perceived application effects, and contextual constraints jointly shape AI adoption and use over time.

To address this gap, this study adopts the Technology–Organization–Environment (TOE) framework and the Technology Acceptance Model (TAM) as complementary analytical perspectives. The TOE framework provides a structural lens for examining the technological conditions, organizational resources, and environmental contexts that shape AI deployment in e-commerce enterprises, while TAM contributes a perceptual lens for understanding how enterprises evaluate the usefulness and usability of AI applications in practice. By integrating these perspectives, this study conceptualizes AI adoption as a multidimensional process shaped by the interaction of technological arrangements, organizational conditions, environmental influences, and enterprise-level perceptions of application effectiveness. Drawing on in-depth interviews with e-commerce enterprises in Anhui Province, this study explores how firms understand, deploy, and adjust AI technologies in real-world business settings. Specifically, the research revolves around the following questions:

1. How are AI technologies currently applied by e-commerce enterprises of different sizes within their business processes, and what are their specific models and technology acquisition channels?

2. How do enterprises perceive the effects of AI technologies in actual operations, including the specific aspects where efficiency or value is demonstrated?

3. What factors influence enterprises' adoption of AI, and what challenges do enterprises face in the process of integrating AI?

4. Looking ahead, what are enterprises' plans, needs, and expectations regarding AI applications?

By systematically examining AI adoption from an enterprise perspective, this study contributes to a more nuanced understanding of digital transformation in e-commerce. It clarifies how AI technologies are operationalized across heterogeneous organizational contexts and offers empirical insights that inform both theory development and practical decision-making for policymakers and technology service providers.

## Literature review

Artificial Intelligence (AI) technology is profoundly transforming the field of e-commerce, with applications becoming increasingly widespread in areas such as marketing, supply chain management, and customer service. In this study, AI specifically refers to digital systems and tools that incorporate machine learning, natural language processing, computer vision, or other algorithmic techniques to support and enhance commercial activities conducted via online platforms. To comprehensively understand the current state of AI applications in e-commerce enterprises, this study synthesizes relevant research findings through a literature review, aiming to provide theoretical support for subsequent empirical research.

### AI adoption in E-commerce

Artificial intelligence technology enables businesses to achieve their goals by accurately analyzing complex and ambiguous external data to uncover commercial value, thereby driving the intelligent development of e-commerce companies [6]. In the context of e-commerce, artificial intelligence primarily refers to the use of systems and tools that incorporate AI technologies or algorithms to support online product or service-related business activities [7]. Currently, AI technology is widely applied across various fields, including image search, chatbots, cybersecurity, data interpretation, predictive analytics, and operations management [1].

AI has been extensively applied across the e-commerce value chain. In marketing and customer interaction, AI-driven content generation, chatbots, and personalized recommendations have significantly enhanced e-commerce performance by improving both customer engagement and conversion rates [4,8]. In e-commerce logistics operations, predictive analytics and intelligent scheduling improve the consistency of the supply chain, as well as the collaboration and coordination of logistics [9]. In security and infrastructure, AI supports cybersecurity systems and real-time threat detection, which is particularly beneficial for SMEs through scalable services such as SecaaS [10]. Furthermore, AI-driven data analytics and decision support systems enable businesses to extract actionable insights and optimize platform strategies [11].

From a functional perspective, AI adoption in e-commerce can be categorized into four key domains: customer engagement (such as virtual assistants), operational optimization (such as demand forecasting), intelligent decision-making (such as pricing and inventory models), and risk management (such as fraud detection) [3,11,12]. Each domain reflects varying levels of technological intensity and organizational adaptation [13]. Moreover, enterprises can implement artificial intelligence through multiple channels, including AI resources embedded within e-commerce platforms, API or module integration, and in-house development [14]. This indicates that AI adoption in e-commerce is not uniform but occurs through diverse configurations and integration pathways.

While AI's presence in e-commerce is increasing, the depth and customization of its adoption exhibit significant variations across different enterprise types and digital ecosystems [15]. Multidisciplinary perspectives emphasize that AI diffusion across enterprises is mediated by organizational readiness, regulatory environments, and ecosystem dependencies, resulting in differentiated adoption trajectories rather than homogeneous implementation outcomes [16]. Large-scale industry surveys similarly report that, although the overall rate of AI adoption continues to increase, the tangible benefits of AI remain concentrated among a limited group of enterprises that are able to scale and integrate AI more deeply into their

operations [17]. In contrast, small and medium-sized enterprises tend to rely on standardized, platform-based, or third-party AI solutions due to constraints in resources, technical expertise, and data infrastructure, resulting in more limited and function-specific applications [18]. Together, these studies highlight that AI adoption in e-commerce is inherently layered, shaped by firm size, digital readiness, and contextual conditions, rather than representing a uniform technological adoption across enterprises.

## TOE & TAM in AI adoption

Current research on organizations adopting artificial intelligence mainly relies on established models such as the Technology-Organization-Environment (TOE) framework and the Technology Acceptance Model (TAM) [13]. The TOE framework provides a macro-level perspective by emphasizing how technological characteristics, organizational readiness, and environmental conditions jointly shape firms' technology adoption decisions, while TAM focuses on the cognitive mechanisms through which users evaluate and accept new technologies. By combining these two perspectives, it is possible to obtain a more comprehensive understanding of AI adoption that incorporates both contextual conditions and evaluative perceptions [19].

The informatization foundation of enterprises is an essential prerequisite for the adoption of artificial intelligence technology [20]. Along with technological capabilities, it determines the depth and breadth of an enterprise's mastery of AI technology and critically influences the complexity of the AI technologies planned for implementation [21]. Furthermore, the maturity of AI technology tools is a necessary condition for encouraging enterprises to adopt AI [22]. The implementation costs and complexity of AI technology directly affect an enterprise's ability to successfully integrate AI into its business functions [23].

Larger organizations possess greater human and financial resources, which often enable them to derive more significant benefits from the use of artificial intelligence technology [20]. Additionally, organizational compatibility plays a crucial role in AI adoption, with flexible organizational adaptability being more conducive to coordinating innovative technologies [19]. Strong leadership support, along with the technological awareness and endorsement of senior management, can substantially accelerate an organization's adoption of artificial intelligence and effectively overcome resistance to technological change [19,24]. Furthermore, internal technical training, project collaboration, and the establishment of dedicated AI teams can reduce implementation challenges and enhance the organization's overall adaptability. [25].

Due to differences in the countries where they are located, technological systems, national innovation systems, and the supportive or obstructive frameworks influencing enterprises' adoption and use of artificial intelligence technology can vary [20]. Given the resource constraints faced by small and medium-sized enterprises, the use of artificial intelligence technology requires additional government support [26]. Additionally, market pressures from competitors and partners, along with business pressures arising from the growing demand for customer personalization, are important external driving forces that determine whether enterprises adopt artificial intelligence technology [27].

Perceived usefulness generally denotes the degree to which the utilization of a specific technology enhances work performance, whereas perceived ease of use relates to how effortless the operation of that technology is perceived to be [28]. Expectations regarding the advantages, anticipated benefits, and convenience offered by artificial intelligence technology can motivate small and medium-sized enterprises to adopt new technological innovations [29]. Moreover, the adoption of artificial intelligence is significantly influenced by perceived usefulness, while perceived ease of use does not play a decisive role [28]. Perceived security in electronic transactions is a direct determinant of technological willingness and the implementation of e-commerce [30]. Additionally, existing compelling business cases serve as credible sources for adopting new technologies, thereby enhancing trust in the effectiveness and reliability of artificial intelligence technology [22].

The TOE framework captures the technological, organizational, and environmental contexts that shape AI adoption, while TAM explains the cognitive mechanisms through which technologies are evaluated and accepted. Although TAM was originally developed for individual users, in organizational settings technology adoption decisions often emerge from

collective managerial evaluations, in which perceived usefulness and ease of use reflect shared organizational judgments rather than individual attitudes. Accordingly, this study integrates TAM into the TOE framework through an organizational perception lens, linking external contextual conditions with enterprises' internal cognitive evaluations and adoption behaviors. TOE is used to structure the technological, organizational, and environmental contexts surrounding AI adoption, whereas TAM is incorporated to interpret how enterprises collectively assess the usefulness and applicability of AI technologies. This integration enables a more nuanced understanding of AI adoption as both a context-dependent and perception-driven organizational process. At the same time, relying on a single framework may provide only a partial explanation for organizational AI adoption, particularly in platform-based and data-intensive environments such as e-commerce. TOE emphasizes structural conditions but does not explicate how enterprises internally interpret technological value, while TAM centers on evaluation mechanisms without modeling organizational resources or environmental constraints. The combined use of TOE and TAM thus offers an analytically balanced approach that connects contextual conditions with organizational-level perceptions, while situating the analysis at the level of enterprise adoption decisions, rather than modeling broader inter-organizational dynamics or long-term capability evolution.

## Challenges and future trends of AI in E-commerce

Artificial intelligence has become an indispensable component of the e-commerce environment [31]. However, the application of AI technology in e-commerce practices still faces both challenges and opportunities [32]. Limited human and financial resources are considered the primary bottlenecks for companies developing AI solutions [33]. Consequently, larger enterprises have a greater advantage in the use of artificial intelligence technology compared to small businesses [20]. Furthermore, critical factors such as the maturity of a company's IT infrastructure, data quality, and internal skill sets directly influence the feasibility of implementing AI technology [22]. Additionally, the knowledge gap resulting from a lack of understanding and awareness contributes to skepticism and resistance toward AI adoption among companies [3].

With the surge in artificial intelligence applications in e-commerce, data privacy, security compliance, and ethical concerns have become critical issues that must be addressed [31]. Data breaches, misuse, and non-compliance with regulations can impede the adoption of artificial intelligence [3]. Additionally, biased data may produce biased outcomes, leading to unfair treatment of customers in e-commerce [3,34]. This not only damages the reputation of businesses but can also result in legal and social responsibility challenges.

By actively addressing these challenges, e-commerce platforms can better establish trust with users and promote a fair and equitable consumption experience [32]. Simultaneously, artificial intelligence offers unique opportunities for new e-commerce markets to transcend traditional retail models [31]. Particularly in areas such as personalized recommendations, intelligent customer service, and supply chain optimization, AI technology is reshaping the competitive landscape of e-commerce [6]. With the increasing prevalence of artificial intelligence and declining costs, the comprehensive adoption of AI will become more widespread and efficient [33]. The future development of e-commerce needs to find a delicate balance between technological innovation and responsible AI usage to ensure the dual realization of commercial value and social responsibility [31].

## Limitations of literature research

Despite extensive research currently being conducted on the application of artificial intelligence in e-commerce, covering various dimensions such as intelligent recommendations, automated customer service, and advertising optimization, the existing literature still shows significant shortcomings in terms of theoretical perspectives, research subjects, and methodological approaches.

From a theoretical perspective, although the TOE and TAM models are widely referenced in research on AI technology adoption, most literature focuses on a single dimension, lacking a systematic understanding that integrates different dimensions and the interplay of multiple factors. This makes it difficult to fully reveal the dynamic interaction logic among

technology, organization, and environment. Regarding research subjects, existing studies predominantly examine AI practices in large enterprises or platform providers, often overlooking the actual application pathways and strategic adjustments of small and medium-sized e-commerce enterprises, which operate under resource constraints and limited capabilities. This leads to a bias in research conclusions, resulting in an "overestimation of adoption levels and underestimation of real challenges." From a methodological perspective, a large number of current studies rely on quantitative models, which can depict trends and tendencies in technology adoption but struggle to capture the detailed challenges and strategies faced by enterprises at the operational level, lacking a practical perspective that is "embedded in processes."

Therefore, the existing literature on the core issue of "how AI is truly deployed in e-commerce processes" continues to reflect a dual bias of "technological optimism" and "operational blind spots." There is an urgent need for supplementary and corrective research that involves various types of enterprises and employs qualitative methods to engage with real-world scenarios. This study aims to address this academic gap by focusing on the frontline perspective of enterprises and exploring the real pathways and evolutionary logic of the integration of AI and e-commerce processes.

## Methodology

This study employs qualitative research methods and develops a theoretical analytical framework based on the Technology-Organization-Environment (TOE) framework and the Technology Acceptance Model (TAM). It takes into account four dimensions: technological characteristics, organizational capabilities, external environment, and user perception, providing a systematic support for explaining enterprises' AI adoption behavior. From a business perspective, the study comprehensively examines the current application status, effects, and future adoption intentions of AI technology among e-commerce companies, aiming to reveal the application characteristics and real challenges of AI technology in the field of e-commerce.

### Sample selection and research context

In the selection of research subjects, this study employs a quota sampling strategy to ensure diversity and analytical comparability across enterprise types. Eight e-commerce companies from Anhui Province were selected, including two large, two medium, two small, and two micro enterprises, enabling systematic comparison of AI application patterns across different organizational scales. The sampled enterprises cover multiple e-commerce formats, including live-streaming e-commerce, domestic platform-based e-commerce, cross-border e-commerce, and hybrid models that combine online and offline operations. They also vary in industry background and operational complexity, reflecting heterogeneous technological contexts, organizational capacities, and environmental conditions. The basic profiles of the interviewed enterprises—including business type, scale, platform orientation, and key characteristics—are summarized in Table 1, providing a structured basis for cross-case analysis.

Anhui Province has experienced rapid growth in its e-commerce sector in recent years, with total e-commerce sales reaching a level above the national average. As a representative province in central China, Anhui reflects both emerging digital transformation dynamics and typical structural constraints faced by many Chinese e-commerce enterprises, thereby providing a meaningful empirical context for exploring AI adoption patterns with broader analytical relevance.

### Interview design and data collection

For data collection, this study utilized semi-structured interviews with leader of e-commerce operations departments in each enterprise. The interviews were designed to capture in-depth insights into enterprises' primary e-commerce platforms, AI applications across core e-commerce processes, perceived effectiveness of AI use, environmental and organizational influences on adoption, and future planning for AI deployment, thereby aligning data collection with the analytical structure of this study.

**Table 1. Profile of Interviewed E-commerce Enterprises.**

| Enterprise Code | Firm Size | Industry | Business Type | Experience (Years) | Key Characteristics |
|---|---|---|---|---|---|
| E1 | Large | E-commerce sales | Cross-border E-commerce | 7 years | Data-intensive operation with multi-platform global reach and high automation demand. |
| E2 | Large | Tea | Live streaming E-commerce | 5 years | With short videos as the orientation and multi-platform live streaming as the core. |
| E3 | Medium | Pharmaceutical | Domestic E-commerce | 4 years | Offline–online integration in a highly regulated industry with multi-platform operations. |
| E4 | Medium | Kitchen Supplies | Hybrid E-commerce | 5 years | Domestic and cross-border retail across multiple platforms in a highly competitive market. |
| E5 | Small | Snack Foods | Domestic E-commerce | 4 years | Multi-store operation on price-sensitive platforms in high-frequency consumer goods. |
| E6 | Small | Building Materials | Cross-border E-commerce | 7 years | Cross-border sales via both independent website and international platforms. |
| E7 | Micro | Thermos Cup | Cross-border E-commerce | 6 years | Vertically integrated production and sales across platforms and independent channels. |
| E8 | Micro | General Merchandise | Domestic E-commerce | 5 years | Small-scale retail across several domestic platforms with platform-dependent operations. |

To ensure the focus and depth of the interview data collection, this study designed and utilized an interview outline consisting of four main questions (Appendix), arranged in a specific order. Each question includes several sub-questions to prevent respondents from straying off-topic and to obtain more structured and informative answers. To ensure theoretical consistency and construct validity, the four main interview questions were mapped to the dimensions of the TOE and TAM frameworks. Specifically, the question on current application status corresponds primarily to the technological context of TOE, exploring the types and integration of AI technologies. The application effectiveness question reflects perceived usefulness and ease of use in TAM, but at the organizational level, focusing on firms' overall evaluations of AI benefits. The influencing factors question links to the organizational and environmental dimensions of TOE, examining internal resources and external pressures. Finally, the future plans and expectations question integrates all three TOE dimensions to explore enterprises' strategic intentions and technology evolution paths.

The interview outline was developed based on the TOE and TAM theoretical frameworks and was refined following a round of pre-test interviews to ensure the clarity and relevance of the questions. Interview questions concerning perceived usefulness and ease of use were designed to capture organizational-level perceptions rather than individual attitudes, thereby aligning TAM constructs with enterprise decision-making processes. The data collection process employed a quota sampling strategy across different enterprise size categories and continued until a preliminary level of theoretical saturation was achieved, with no substantially new themes emerging in the final interviews. This indicated that the key constructs related to AI adoption in e-commerce had been thoroughly explored within the scope of the study.

## Ethical considerations

All interview procedures complied with established ethical standards for social science research. Prior to data collection, each participant was informed of the study's purpose, the voluntary nature of participation, and the confidentiality and anonymization of all responses. Verbal informed consent was obtained from all participants before the interviews, which were conducted between March and July 2025. Consent was documented by the interviewer through audio recording at the beginning of each interview and noted in the interview records.

No personal identifiers or sensitive individual information were collected, and all transcripts were anonymized at the enterprise and individual levels. The study involved professional interviews with enterprise representatives and did not

collect personal, medical, or vulnerable population data. Based on institutional guidelines, this type of research did not require formal institutional review board (IRB) approval. All data were stored securely and used solely for academic research purposes.

## Data analysis and research rigor

The coding process followed a thematic analysis approach using a hybrid strategy that integrated inductive and deductive procedures [35]. In the initial stage, open coding was conducted across all interview transcripts to identify recurring concepts, practices, and expressions related to AI adoption in e-commerce operations. These initial codes were derived directly from the data and reflected participants' descriptions of platforms, applications, perceived effects, and contextual influences. At this stage, no predefined categories were imposed, allowing patterns such as "platform reliance," "manual verification," and "tool integration pathways" to emerge from participants' own language.

In the subsequent stage, related codes were systematically organized into higher-order themes through axial coding, enabling the consolidation of patterns across enterprises of different sizes. During this phase, the TOE framework and TAM were introduced as sensitizing concepts rather than rigid coding templates [36]. Emerging themes were iteratively compared with the dimensions of Technology, Organization, Environment, and Perceived Usefulness to examine how empirically grounded patterns aligned with, extended, or nuanced these theoretical constructs. For example, inductively generated codes related to "free platform tools," "paid plug-ins," and "in-house systems" were synthesized into the theme "application pathways," which was then analytically linked to the technological and organizational dimensions of TOE.

To enhance analytical rigor and minimize confirmation bias, several measures were implemented. First, coding decisions were continuously compared across cases and enterprise sizes to identify convergent and divergent patterns. Themes were retained only when supported by evidence from multiple enterprises, while boundary cases and contradictory views were explicitly examined to refine category definitions. Second, a member-checking procedure was applied, whereby interview summaries were returned to selected participants for confirmation of factual accuracy. Third, a second researcher conducted an independent review of the coding structure and thematic interpretations.

Through these procedures, the study sought to improve the credibility, dependability, and transparency of the qualitative analysis. The final coding structure and thematic framework derived from NVivo analysis are presented in Fig 1.

## Interview data findings & analysis

To maintain the focus of the research and facilitate a structured analysis and comparison of interview data, this study categorizes and organizes the interview content by theme. These themes primarily include: the main e-commerce platforms used by the enterprises; specific applications of artificial intelligence technology in e-commerce processes; pathways and modes of AI implementation; organizational influencing factors; external environmental factors; the actual effects following AI adoption; and the enterprises' plans for future AI applications. Based on these themes, this article systematically summarizes and analyzes the interview data (Table 2).

## The main platform of enterprise e-commerce

The interview results indicate that e-commerce enterprises of different sizes exhibit diverse characteristics in platform selection, involving various categories such as traditional e-commerce, cross-border e-commerce, mobile e-commerce, and social/live-streaming e-commerce. Regarding traditional e-commerce platforms, companies typically choose mainstream B2C and C2C platforms like Taobao, Tmall, JD.com, and Pinduoduo. In cross-border e-commerce, some companies opt for platforms like Amazon, Temu, Shopify independent sites, and Alibaba International Site, which primarily target the global market. For mobile e-commerce, some enterprises utilize Alipay Mini-Program Engine stores to establish their online sales channels. Additionally, social and live-streaming e-commerce have become significant channels

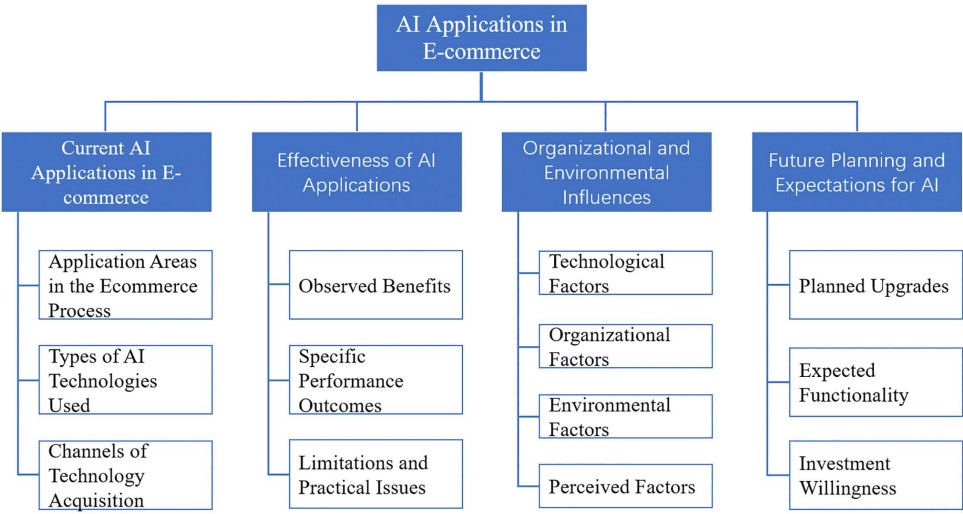

**Fig 1. Structure for Thematic Codes from Interview Data in Nvivo15.**

for an increasing number of companies. Some businesses actively engage on short video and live-streaming platforms such as Douyin, Kuaishou, and WeChat Channel, Channels, content marketing and interactive experiences to enhance consumer purchasing intentions. Notably, instant retail platforms like Meituan and Ele.me have also appeared appeared the operational scope of these enterprises, indicating that some e-commerce companies are exploring localized O2O models to enhance enhance experience through instant delivery and meet the demand for fast consumption.

## AI applications in E-commerce processes

Artificial intelligence technology has been widely applied across multiple core areas of e-commerce businesses, primarily including product content generation and optimization, advertising placement, data analysis, customer service automation, logistics management, and others (Fig 2). As shown in Fig 2, the values indicate how many enterprises reported using AI in each domain. Since individual firms typically adopt AI across multiple functions, these frequencies reflect the relative prevalence of different application areas. Companies generally utilize AI to optimize product titles, descriptions, images, and videos, thereby enhancing content quality and appeal. Additionally, advertising placement has become a key area for AI applications, which not only includes automated placement and intelligent matching but also involves precise traffic analysis to help businesses optimize promotional effectiveness. Data analysis is an important support for AI applications, encompassing operational data, advertising keywords, traffic sources, and competitor analysis, providing data-driven decision support for businesses to formulate operational strategies. Companies also employ AI-powered customer service systems to achieve automatic responses, monitor return rates, and facilitate intelligent interactions, thereby improving customer service efficiency. In logistics management, AI technology supports inventory management, smart delivery, and anomaly detection. Some companies use AI to monitor data on competitors' platforms, allowing them to promptly adjust their operational strategies. Overall, AI technology has become deeply integrated into the entire e-commerce process, with companies of various sizes are exploring more efficient and intelligent AI application models to improve operational efficiency and competitive advantage.

## Application pathways and modes of AI use

Large enterprises typically construct multi-layered application architectures that combine platform-native AI functions, third-party tools, and internally developed systems. This hybrid mode allows them to customize algorithms, integrate

**Table 2. Interview Quotes Supporting AI Adoption Analysis in E-commerce.**

| Theme | Supportive quotes |
|---|---|
| **Main Platforms** | 'Online, we have five major sections: Douyin, JD, PDD, Kuaishou, and WeChat Channel. (Interviewee #E2)', 'We mainly engage in online cross-border business, with two platforms: an independent site and Alibaba International Site. (Interviewee #E6)' |
| **AI Application Areas** | 'This section mainly involves copywriting, as we need to create some operational titles, subtitles, and marketing phrases. Additionally, we will write some advertising copy during the advertising flow process. (Interviewee #E8)', 'The keywords for the advertising product, and the specific amount of money to invest, will have a suggestion on the platform that you can follow for your investment. It will also analyze which advertising phrases for this product might be better. (Interviewee #E7)', 'Sometimes I use AI to send my requests to it, asking it to help me filter out the key information and output the table for me. (Interviewee #E5)' |
| **Application Pathways and Modes** | 'Besides using the systems like Amazon's backend and DeepSeek for basic data analysis, our company has also developed our own tool specifically for optimizing the display of Amazon front-end. (Interviewee #E1)', 'Besides the AI provided by Amazon, we also use SellerSprite and Lingxing. These two aren't built into the platform, so we have to pay to use them. (Interviewee #E4)', 'Other than third-party free AIs like DeepSeek, all the AI tools we use are built-in platform features. (Interviewee #E5)', 'Except for the platform's ad placement, all the other tools we use right now is basically free. (Interviewee #E8)' 'For things like product titles and descriptions, we run them through DeepSeek to translate them into the local language. The error rate is actually pretty low, but we still have someone double-check everything manually, just in case. (Interviewee #E1)', 'The platform provides us with the keywords and the exact budget needed for ad placement. and we'll sometimes just follow those recommendations when we launch a campaign. (Interviewee #E7)' |
| **Observed Effects** | 'Amazon has its own built-in AI, which I have used before for generating images. You input a command, and it produces images quite quickly. I feel that the calculations it makes seem a bit too rough, the profits they calculate are higher than I expected. It will be much better than the words you find yourself because the words they recommend have a high level of relevance. (Interviewee # E7)', 'During the promotion process, sales and conversion rates increased by about 10%, but it is not very high. (Interviewee # E5)', 'It can provide us with a framework that is only for reference and not operational. Ultimately, it still relies on our sales personnel to do the advertising themselves. […] these market analysis tools are quite effective. (Interviewee # E1)', 'I think it will help reduce the workload a bit. However, some actual ideas or product choices, as well as certain marketing techniques, still require human involvement. (Interviewee # E6)' |
| **Environmental and Organizational Influences** | 'I think the biggest issue with using AI is its tuning. You need to fine-tune it accurately, feed it the right terms, and getting it to produce the correct video takes a considerable amount of time. Additionally, the learning curve is quite steep; some aspects may involve coding issues. It requires a lot of time to learn how to use it and how to operate it. (Interviewee # E2)', 'Our current advertising data monitoring relies entirely on manual efforts, basically requiring us to keep an eye on it during working hours. […] we have not yet looked into any policies from the government or industry associations. (Interviewee # E8)', 'The management has not considered artificial intelligence at all. […] they are aware of its existence, but it is not included in the company's strategic planning. (Interviewee # E3)', 'Sales satisfaction mainly depends on your product and its quality. […] as far as I know, I haven't heard of any policies or subsidies related to this. (Interviewee # E6)' |
| **Future Plans for AI** | 'I hope it can make further progress in that image processing, helping me achieve things that some graphic designers can't. The program doesn't need to be too complicated; it should be simple and fast, allowing me to use it easily and quickly see results. (Interviewee # E6)', 'What we are looking forward to is that he can quickly help us handle some matters, optimize some of our operational processes, simplify things a bit, and then when we use them, they can bring us better results. (Interviewee # E5)', 'They will consider costs and also the issue of waiting and observing. First, they might look at some other companies, which may yield better operational results, and then they will consider integrating. (Interviewee # E7)' |

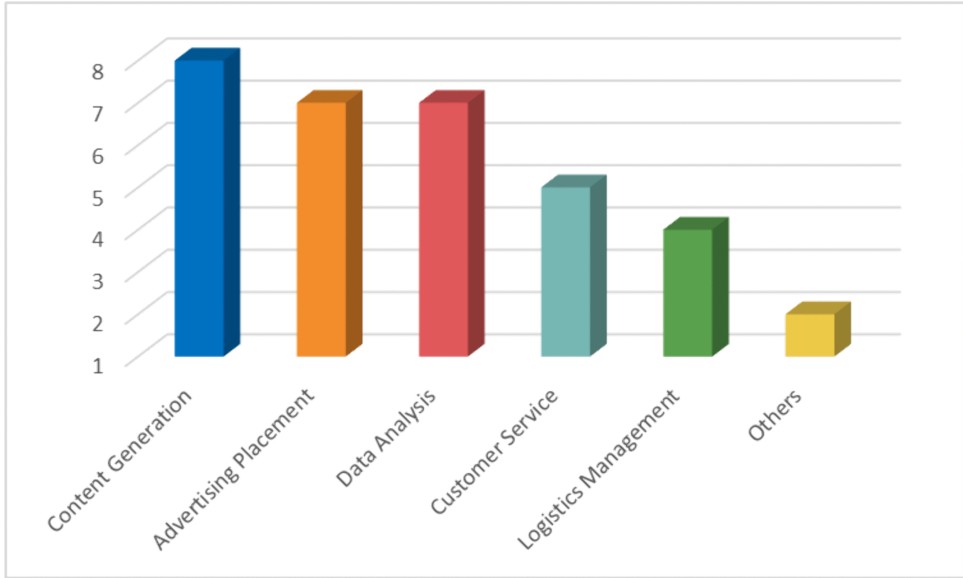

**Fig 2. The main application areas of artificial intelligence in the e-commerce process.**

cross-platform data, and embed AI deeply into decision-making and operational workflows. In contrast, small and micro enterprises predominantly rely on standardized AI services embedded within e-commerce platforms, such as intelligent pricing, keyword optimization, and traffic recommendation tools. These "platform-bound" pathways lower technical and financial barriers, enabling rapid adoption but limiting the depth of customization. Medium-sized enterprises occupy an intermediate position: while they remain dependent on platform ecosystems, they demonstrate a greater willingness and capacity to invest in paid external tools. Across cases, AI deployment follows a platform-centered integration pattern complemented by external tools. Enterprises of different sizes adopt free services, paid solutions, or self-developed systems according to their resource endowments, resulting in a stratified and dynamically adaptive adoption process.

Alongside these differentiated integration pathways, enterprises consistently described a human–machine collaboration mode characterized as "AI pre-processing + human fine-tuning." AI is widely used to perform repetitive and standardized tasks, including drafting product descriptions, generating advertising materials, providing first-round customer responses, and producing analytical reports. Human operators then refine outputs, ensure contextual relevance, and conduct judgment-intensive activities such as compliance checks, product selection, and strategic adjustments. This collaborative configuration allows firms to improve efficiency while maintaining control over quality and risk. Rather than replacing human labor, AI functions as an augmentative layer embedded within existing workflows, reshaping task boundaries and redistributing cognitive effort.

## Effectiveness of AI applications

The application of artificial intelligence technology in e-commerce companies has generally enhanced work efficiency and reduced operational costs, though certain limitations remain. In content generation, AI can accelerate the production of copywriting, images, and videos, thereby reducing human labor. This capability has been particularly valuable, especially in situations with limited human resources. However, the accuracy, creativity, and compliance of AI-generated content still require improvement. Companies generally report that AI-generated copywriting may not fully or accurately describe products and may even risk violating advertising regulations, necessitating human review. Regarding advertising placement, AI technology has optimized placement strategies, improving the advertising production ratio and making promotional

expenses more cost-effective. Nevertheless, some companies have noted that the advertising optimization suggestions provided by AI remain at a conceptual level and lack strong practical applicability, ultimately still requiring human decision-making and execution.

In the fields of data analysis and customer service automation, AI technology offers businesses more systematic and efficient analytical support, such as automatically organizing reports, optimizing advertising keywords, and analyzing market trends, making operational management more precise. However, some companies have pointed out that AI analysis may deviate from actual conditions, particularly in areas like profit estimation and customer satisfaction assessment, which still require human judgment. While automated customer service responses have increased response speed, many businesses believe that AI-driven customer service still requires human assistance and cannot fully replace human agents. In logistics management, AI's capabilities in inventory control, delivery optimization, and anomaly detection help companies improve supply chain efficiency. Overall, the application of AI in e-commerce has brought significant improvements in efficiency and cost optimization, but there remains potential for further advancements in the areas of content quality, advertising optimization, and data accuracy. Companies still need to combine human intervention to fully leverage the advantages of AI technology.

## Environmental and organizational influences

In the process of applying artificial intelligence technology, the skill level of personnel within e-commerce companies is crucial for the effective use of AI. Many companies emphasize that employees' proficiency with AI tools directly impacts outcomes such as writing instructions, adjusting parameters, and analyzing data. Additionally, most companies still rely on manual review processes, especially in areas like content generation, advertising placement, and customer service responses, to ensure accuracy and compliance. Furthermore, the attitudes and decisions of company managers also influence the depth of AI usage; some leaders adopt a wait-and-see approach or lack sufficient knowledge support, which hinders technology adoption. At the same time, many companies believe that the quality of the product itself and the service experience are the primary drivers of sales growth, leading to a relatively cautious investment in AI. Although some companies support the use of AI, they tend to maintain traditional operational frameworks and prefer to utilize AI as an auxiliary tool rather than a complete replacement for human labor.

At the external environment level, the surveyed companies generally showed little consideration for the impact of national or industry policies on the implementation of artificial intelligence applications. This suggests that, at this stage, policy-driven incentives for corporate AI adoption are relatively weak. In addition, some companies refer to the AI usage of their peers in the same industry and adopt a cautious attitude towards AI applications, preferring to observe the AI application results of their competitors before deciding whether to invest further. Consequently, current AI applications in the e-commerce industry are primarily driven by companies' operational needs and internal management decisions, with external environmental factors exerting relatively limited influence.

## Future planning for AI in E-commerce

Although most companies do not yet have clear investment plans for artificial intelligence technology, they have expressed potential needs and expectations for specific functions and tools. Generally, companies hope that AI can play a greater role in enhancing the intelligence of customer service, optimizing the quality of content generation, improving the depth and accuracy of data analysis, and achieving system integration across platforms. Many respondents mentioned that current AI tools mostly provide framework suggestions, lacking specificity and operability, and they hope for more constructive analysis and actionable recommendations in the future. Additionally, companies expect AI to provide more efficient and precise support tools at critical operational points such as product selection, marketing strategy optimization, customer relationship management, and customer feedback management, thereby truly achieving cost reduction and efficiency improvements.

The future planning and investment in AI technology by enterprises are influenced by multiple factors, including business strategy, financial capacity, and industry characteristics. Some large companies have regarded AI applications as an important direction for the future and plan to strengthen development in areas such as smart logistics, market analysis, and data integration, although they also express some uncertainty. Meanwhile, small and medium-sized enterprises tend to gradually integrate AI functions based on their existing foundations. Most of these companies are in a state of "observing the effects on peers and assessing cost-effectiveness," planning to cautiously advance the application of AI technology according to their own development stage and resource availability.

## Discussion & comparative analysis

Based on the analysis of interview data, this article further reveals the similarities and differences between the practical applications of AI in enterprises and existing research. Through comparative analysis, it clarifies the specific contributions of this study in terms of theoretical validation and empirical supplementation, and explores the structural challenges and transformation pathways of AI empowerment in e-commerce.

### Summary of key findings

(1) The diversification of platform operations drives the differentiation of AI application demands.

The diversification of platform operations in e-commerce enterprises has driven differentiated demands for AI technology applications. The operational logic, content standards, and marketing mechanisms established by various platforms form the foundation for the distinct application of artificial intelligence tools by enterprises. Cross-border e-commerce platforms such as Amazon and Shopify emphasize language adaptation, content standardization, and precise advertising matching, leading enterprises to rely more heavily on AI for multilingual copywriting, image and text generation optimization, and automated advertising recommendations. In contrast, domestic comprehensive platforms like Taobao, JD.com, and Pinduoduo focus on differentiated product presentation and precise recommendations driven by big data, prompting enterprises to prefer using AI to assist with content optimization, keyword matching, and operational data analysis within platform rules. With the rise of live-streaming e-commerce platforms such as Douyin and Kuaishou, some enterprises are increasingly focusing on AI-driven video generation and optimization to enhance traffic acquisition and conversion efficiency. The diversification of platform operations not only broadens the application scenarios for AI but also profoundly influences enterprises' decision-making processes regarding the selection, deployment, and investment strategies for AI tools.

(2) AI applications cover the entire chain of the e-commerce process, but the depth and customization still need to be improved

Currently, e-commerce companies have widely adopted artificial intelligence technologies across various domains, including product content optimization, advertising placement, customer service, data analysis, and logistics management. This has enabled preliminary full-chain technical coverage of e-commerce operations. Especially in high-frequency tasks like product copy generation, image processing, and intelligent customer service, AI tools have significantly improved efficiency and reduced the burden on human resources, becoming an important auxiliary force in daily operations. However, in practice, the use of AI tools still faces challenges related to insufficient depth and limited customization. Many respondents reported that existing AI tools can offer general framework suggestions but lack specificity, detail, and platform adaptability, often requiring substantial manual modifications and reviews, particularly in meeting platform content standards, user personalization requirements, and industry-specific regulations, which AI remains unable to fully deliver. This disconnect between technology capabilities and practical demands restricts the evolution of AI from being merely a "tool" to "decision support" or "process replacement."

Especially for small and medium-sized enterprises, the high costs of technology development, the complexity of tool usage, and the lack of technical support teams often limit them to using only open-source or free tools, which offer restricted functionality. This makes it challenging for SMEs to systematically integrate and customize solutions according to their own business processes. This situation of "technology existing but difficult to align with the capabilities and scenarios of enterprises" leads SMEs remaining at a superficial level of AI usage, unable to achieve genuine business upgrades driven by AI. Although AI has initially been recognized for its perceived ease of use and usefulness, the gap between actual user experience and expected benefits affects the likelihood of its continued adoption and expanded application. Therefore, to promote high-quality AI applications in e-commerce, it is necessary not only to continuously optimize the technology itself but also to focus on building a matching mechanism between personalized enterprise scenarios and technical solutions.

(3) The effects of AI applications are beginning to emerge, but they still depend on human collaboration and supervision

E-commerce companies generally believe that current AI applications have played a preliminary role in improving operational efficiency, content generation, advertising placement, and customer service. However, these companies emphasize that the application of AI has limitations and dependencies: whether it is the accuracy of generated content, the practicality of advertising placement suggestions, or the intelligence of customer service responses, the results produced by AI still face challenges such as generalization, lack of contextual understanding, and an inability to fully meet actual needs. This situation means that AI functions more as a "collaborative tool" rather than a "replacement tool," requiring human cooperation, review, and correction.

The mechanism of "human-machine collaboration" reflects that current AI technology has not yet matured to the level of fully autonomously executing complex e-commerce tasks. Companies typically implement manual intervention processes at critical stages such as content review, customer interaction, and strategy execution to avoid risks such as low-quality AI-generated content, platform violations, and misinformation. Especially in areas like compliance with advertising laws, professional product descriptions, and diverse customer communications, AI still lacks the ability for contextual judgment and logical reasoning, requiring human judgment to achieve "precise output." Additionally, many small and medium-sized enterprises, constrained by limited technical capabilities and understanding of platform regulations, must rely on experience and manual judgment to verify AI-generated suggestions, further increasing the workload on personnel (Fig 3). Overall, the application of AI in e-commerce remains at the "assisted intelligence" stage, and its collaboration with humans remains a key guarantee for achieving actual results.

(4) Organizational capability and AI literacy within enterprises have become key factors driving adoption, whereas external policy support remains insufficient

Although some enterprises have introduced AI tools, limited employee capability in using these technologies and insufficient managerial understanding often restrict AI to superficial applications, preventing it from delivering meaningful

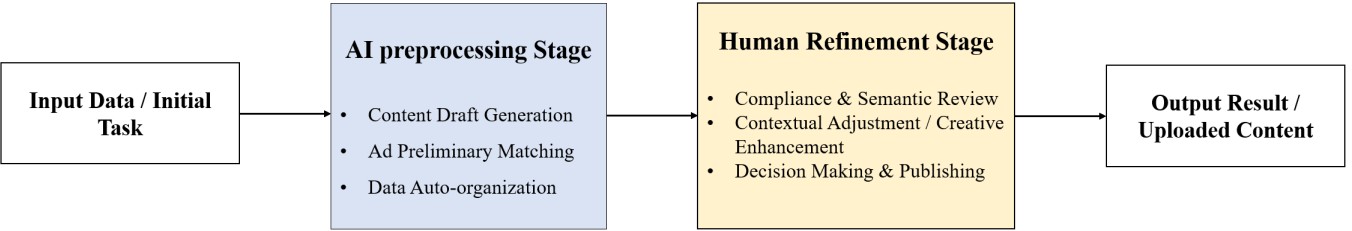

**Fig 3. AI-Human Collaborative Workflow in E-commerce.**

organizational empowerment. As a result, the functions of AI are confined to superficial applications and fail to achieve genuine empowerment. Additionally, the "default templated output" generated by AI systems requires employees to exercise judgment and optimization skills; without these, there is a risk of producing outputs that do not align with the company's product style or platform rules. This disparity in AI literacy significantly influences the company's perception of AI's usefulness and its willingness to adopt the technology.

At the same time, most enterprises demonstrate a lack of attention to policy guidance and industry support when assessing the external environment. Almost all interviewed companies did not proactively mention national or local policies promoting AI + e-commerce, nor did they consider policy dividends as an external driving force for AI adoption. This phenomenon reveals a weakening of the environmental dimension within the TOE framework and indicates that current AI adoption primarily depends on the internal capabilities of enterprises rather than external drivers. It also suggests that policymakers, in promoting the integration of "AI + e-commerce" in the future, can transform the external environment from "potential" to "explicit supportive force" through demonstration projects, policy guidance, and other strategies.

(5) The deployment of AI technology demonstrates a dynamic adaptation mechanism and a paradigm of human-machine collaboration

The current integration of AI technology within e-commerce processes exhibits significant dynamic layering characteristics and human-machine collaboration models. As illustrated in Fig 4, this study anchors AI adoption in a platform-centered architecture supplemented by external tools. However, enterprises follow differentiated paths according to their scale, technological capacity, and resource endowment. Micro and small firms display largely similar patterns, relying primarily

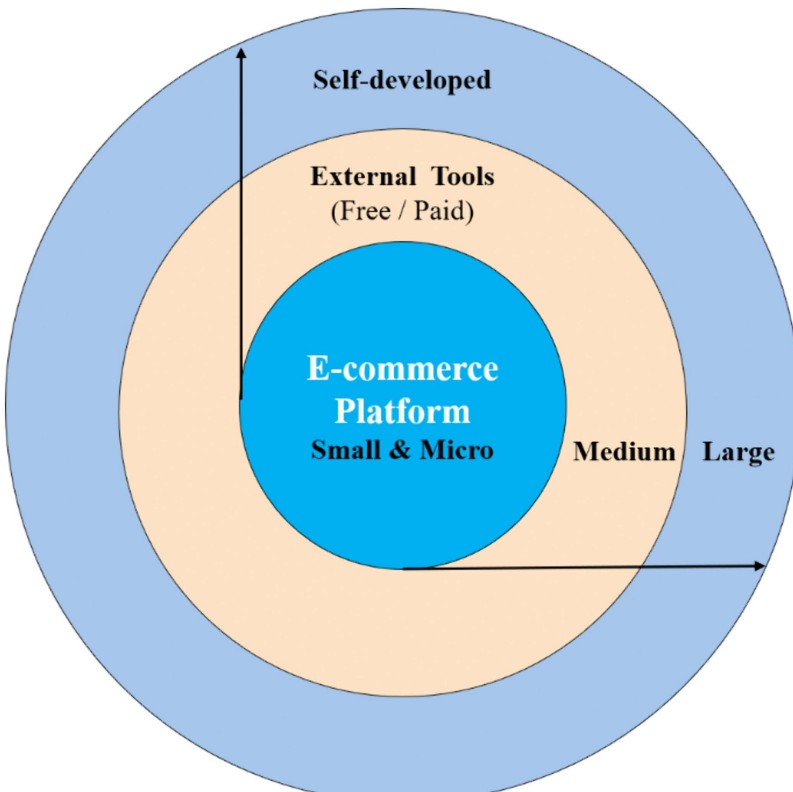

**Fig 4. Enterprise AI Integration Model in E-commerce.**

on platform-embedded AI functions and a limited number of low-cost or free tools. Medium-sized enterprises, while still operating within a "platform + third-party tools" structure, demonstrate a higher level of investment in paid AI services and cross-platform management systems, reflecting a transitional stage of capability expansion. Large enterprises further extend this architecture by incorporating self-developed AI solutions, forming a three-layer structure of "platform core + external tools + self-developed" that enables deeper integration and refined operational control. This stratified deployment challenges the traditional "one-size-fits-all" assumption of technology adoption. Instead, it suggests that the depth and form of AI integration in e-commerce are shaped by enterprises' evolving resource conditions and organizational capacities and are better understood as a process of personalized and adaptive alignment rather than generalized application.

Simultaneously, the application of AI technology is driving the re-engineering and reconstruction of e-commerce processes. Companies are generally developing a collaborative work paradigm of "AI pre-processing + human fine-tuning." In areas such as drafting initial copy, designing advertising materials, and providing rapid customer service responses, AI takes on standardized and repetitive tasks, while humans focus on high-value creative work, including product selection and compliance judgment. This model not only enhances operational efficiency but also significantly reduces labor costs, all while ensuring the professionalism and compliance of the content. Therefore, the integration of AI is not merely a replacement through automation, but rather a symbiotic mechanism that complements human capabilities, demonstrating a dual-track collaborative approach of technology embedding and human support in practical operations.

## Comparative analysis: Interview findings vs. literature review

This study compares firsthand interview data with existing literature, not only validating the applicability of certain theoretical perspectives but also revealing the limitations of current research when confronted with real-world complexities.

In terms of theoretical validation, this study confirms the fundamental effectiveness of the TOE and TAM theoretical frameworks in explaining the adoption behavior of AI technology within enterprises. Feedback from company interviews regarding the "perceived usefulness" and "ease of use" of artificial intelligence closely aligns with the core concepts of the TAM model. Simultaneously, variations in companies' technological adaptability, internal organizational capabilities, and perceptions of the external environment also resonate with the core logic of the three dimensions—"Technology-Organization-Environment"—in the TOE model. This confirms that applying these two models in the e-commerce context has a robust theoretical foundation and strong explanatory power.

However, existing literature often assumes that AI technology is a relatively standardized and replicable technical solution applicable to all types of enterprises [37,38]. This study finds that, in practice, there are significant layered integration patterns in AI applications. Large enterprises tend to build a composite technical system of "platform core + external tools + self-developed" solutions, whereas small and medium-sized enterprises typically adopt a "platform-bound" lightweight integration model. This distinction challenges the traditional "one-size-fits-all" adoption assumption in the literature and suggests that researchers should thoroughly consider the actual disparities in organizational capabilities and digital infrastructure when exploring technology adoption.

Although the literature generally emphasizes the important role of external environmental factors, such as policy guidance and industry pressure, in technology diffusion [26,28], interviews revealed that most of the interviewed companies have very limited attention to national or industry policies, and the actual impact of institutional factors is often overestimated. This finding suggests that the current institutional environment's incentive mechanisms for promoting AI applications are not yet fully effective, and there may be biases in how policy variables are represented in research. Future studies should focus more on the institutional signals that companies "perceive" rather than on the "existing" institutional frameworks.

Finally, in terms of the collaboration mechanism between AI and human labor, this study proposes a collaborative paradigm of "AI preprocessing + human fine-tuning," revealing the realistic path where AI is more likely to serve as an enhancement tool rather than a complete replacement within e-commerce enterprises. This finding offers a significant response

to the prevailing "substitutive narrative" [39,40] and serves as a foundational reference for constructing more practical AI deployment models in the future.

## Theoretical implications

First, while the TOE and TAM frameworks provide a valid foundation for understanding enterprise-level AI adoption, our findings suggest the need to move beyond static constructs. The evidence of differentiated AI integration paths—namely, "platform-bound adoption," "external tool enhancement," and "autonomous system development"—indicates that technology assimilation is a dynamic, staged process. Therefore, this research constructs a "Enterprise AI Integration Model in E-commerce" revealing the phased and differential characteristics of enterprises in technology deployment. This model dynamically supplements the "technological dimension" in TOE theory, emphasizing the non-linearity of the technology adoption process and the matching of organizational conditions.

Second, our discovery of the "AI pre-processing + human refinement" collaboration model introduces a complementary human-machine interaction paradigm. Rather than viewing AI as a replacement, our findings support the concept of AI as an augmentative partner. This contributes to recent theoretical shifts toward co-adaptive technology use, in which AI and human labor are jointly configured based on task complexity and risk. It also highlights the importance of micro-level coordination mechanisms, which have been underexplored in current technology adoption theories.

## Conclusion

This study is based on the TOE framework and the TAM theory. Through interviews and research on e-commerce enterprises of different sizes, it systematically outlines the current application status of artificial intelligence technology in e-commerce processes, evaluates its effects, examines the environmental and organizational influencing factors, and discusses future development plans. The aim is to reveal the true nature of the integration of AI technology with e-commerce operations and the challenges it faces.

Research has found that artificial intelligence has gradually been integrated into various processes within e-commerce companies, including product content generation, advertising placement, customer service automation, and data analysis, resulting in a relatively complete coverage of the value chain. However, there are still significant shortcomings in depth and customization. Small and micro enterprises, constrained by costs and technical barriers, primarily adopt a platform-bound integration approach, lacking autonomy and flexibility. AI technology has shown initial effectiveness in improving efficiency and reducing human workload, but most applications still require human collaboration and subsequent review, revealing a misalignment between current technological capabilities and the actual operational needs of enterprises. At the organizational level, the AI literacy of employees and the willingness to adopt AI within the organization are key factors determining the effectiveness of AI deployment. Additionally, the lack of policy support and industry guidance in the external environment has led most enterprises to pay little attention to policy information or lack channels to access it, further affecting the acceptance of AI technology. Although large enterprises demonstrate strong strategic integration capabilities, small and micro enterprises tend to adopt a more cautious, wait-and-see attitude. However, research indicates that various types of enterprises generally hold an optimistic outlook on the future development of AI, particularly expressing strong potential demand in areas such as automation, intelligent recommendations, and data-driven decision-making.

In summary, the application of artificial intelligence in e-commerce is currently in a transitional phase of "from point to surface, from shallow to deep." Enterprises exhibit distinct characteristics of layered integration and dynamic evolution in their acceptance and deployment of AI. In the future, it is necessary to further enhance the adaptability and usability of AI technologies in small and medium-sized enterprises, promote the implementation of supportive policies and industry guidelines, and improve AI awareness and operational capabilities within organizations through targeted training. These efforts will unlock greater potential for AI to empower high-quality development in e-commerce.

 

At the same time, this study has several limitations. First, the empirical evidence is drawn from a small number of enterprises within a single province, which may constrain the generalizability of the findings across different regions and institutional contexts. Second, as a qualitative study, the analysis relies on self-reported perceptions and experiences, which may be subject to recall bias or subjective interpretation. Third, the cross-sectional design captures AI adoption at a particular stage and cannot fully reflect its longitudinal evolution.

Future research could address these limitations by expanding the sample across multiple regions or countries, incorporating quantitative methods to test the proposed patterns and mechanisms, and adopting longitudinal designs to track the dynamic evolution of AI adoption over time. Additionally, further studies could examine sector-specific differences and explore how policy instruments and platform governance structures influence organizational AI trajectories. These efforts would deepen our understanding of AI-driven transformation in e-commerce and strengthen the theoretical and practical contributions of this research field.

## Supporting information

**S1 File. Interview questions.**
(DOCX)

## Acknowledgments

None.

## Author contributions

**Supervision:** Mohd Zaidi Abd Rozan.

**Writing – original draft:** Tong Zhu.

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
