## [Decision Letter · Decision Letter 0]

6 Jan 2026

Dear Dr. Zhu,

Thank you for submitting your manuscript to PLOS ONE. After careful consideration, we feel that it has merit but does not fully meet PLOS ONE’s publication criteria as it currently stands. Therefore, we invite you to submit a revised version of the manuscript that addresses the points raised during the review process.

I recommend that it should be revised taking into account the changes requested by the reviewers. Since the requested changes include valuable and constructive reviews, I would like to give you a chance to revise your manuscript. The revised manuscript will undergo the next round of review by same reviewers.

We look forward to receiving your revised manuscript.

Kind regards,

Baogui Xin, Ph.D.

Academic Editor

PLOS One

Journal Requirements:

2. In the ethics statement in the Methods, you have specified that verbal consent was obtained. Please provide additional details regarding how this consent was documented and witnessed, and state whether this was approved by the IRB.

[This research is supported by the project “Application Research of Artificial Intelligence Technology in e-Commerce Enterprises in Anhui Province”, with Fund Number 2024AH052502.].

[This research is supported by the project “Application Research of Artificial Intelligence Technology in e-Commerce Enterprises in Anhui Province ”, with Fund Number 2024AH052502.]

[This research is supported by the project “Application Research of Artificial Intelligence Technology in e-Commerce Enterprises in Anhui Province”, with Fund Number 2024AH052502.]

5. We note that your Data Availability Statement is currently as follows: [All relevant data are within the manuscript and its Supporting Information files]

If your submission does not contain these data, please either upload them as Supporting Information files or deposit them to a stable, public repository and provide us with the relevant URLs, DOIs, or accession numbers. For a list of recommended repositories, please see https://journals.plos.org/plosone/s/recommended-repositories

7. Your ethics statement should only appear in the Methods section of your manuscript. If your ethics statement is written in any section besides the Methods, please move it to the Methods section and delete it from any other section. Please ensure that your ethics statement is included in your manuscript, as the ethics statement entered into the online submission form will not be published alongside your manuscript.

Reviewers' comments:

Reviewer's Responses to Questions

**Comments to the Author**

1. Is the manuscript technically sound, and do the data support the conclusions?

Reviewer #1: Partly

Reviewer #2: Yes

2. Has the statistical analysis been performed appropriately and rigorously?

Reviewer #1: No

Reviewer #2: Yes

3. Have the authors made all data underlying the findings in their manuscript fully available?

Reviewer #1: No

Reviewer #2: Yes

4. Is the manuscript presented in an intelligible fashion and written in standard English?

Reviewer #1: No

Reviewer #2: Yes

Reviewer #1: Thank you for the opportunity to review this manuscript. Overall, the topic is timely and the study is potentially suitable for PLOS ONE, but my answers above reflect several areas where the reporting is not yet sufficiently clear to allow a confident judgement of technical rigor.

Regarding technical soundness and whether the data support the conclusions, the manuscript presents relevant interview evidence and includes useful quotations, but some higher-level claims and the proposed conceptual framing would be stronger if the link between the interview data and the conclusions were more explicit and consistently demonstrated across cases. At present, a number of interpretations appear reasonable, yet readers cannot always see a clear audit trail showing how specific themes and model components were derived from the coding process and supported across multiple enterprises, including any boundary cases or divergent views.

For the question on statistical analysis, my response reflects that this is primarily a qualitative interview study rather than a quantitative one. In this context, “rigor” depends on transparent and well-justified qualitative analytic procedures. The manuscript would benefit from a fuller description of how thematic analysis was conducted (e.g., whether coding was inductive, deductive, or hybrid), how the TOE and TAM frameworks were applied (as sensitizing concepts versus a formal coding frame), and what steps were taken to minimize confirmation bias and ensure analytic trustworthiness. The current level of detail is not yet sufficient for readers to evaluate the robustness of the analysis.

For data availability, the manuscript’s statements are not fully consistent and do not clearly indicate what underlying interview materials are available. PLOS ONE expects that data underlying the findings are accessible unless there is a justified ethical or legal restriction. For interview-based research, this typically requires either anonymised/appropriately redacted transcripts or a clearly described controlled-access pathway if public sharing is not possible. As currently written, it is difficult to determine what, if any, underlying data can be accessed to verify the findings.

On research ethics, the reporting also requires clarification. The manuscript notes verbal informed consent and suggests formal institutional ethics approval was not required, yet the submission information appears to indicate otherwise. Because this work involves human participants, the ethics pathway should be stated unambiguously, including whether there was ethics approval, exemption, or waiver, and how consent, confidentiality, data handling, and anonymisation were managed. Presenting these details clearly within the Methods section (not only in Declarations) would improve transparency and compliance with journal expectations.

Finally, the manuscript is generally intelligible in English, but it would benefit from careful language polishing to reduce ambiguity and improve precision. In addition, conclusions that suggest broader representativeness should be moderated given the small sample and single-province setting, and any quantitative-sounding improvements (e.g., percentage changes) should be clearly framed as participants’ estimates unless independently verified.

I did not identify clear evidence of dual publication from the materials available to me, but I recommend that the authors ensure the manuscript and any related outputs (e.g., conference proceedings, reports, or preprints) are appropriately disclosed according to journal policy. Overall, addressing the points above would significantly strengthen the manuscript’s methodological transparency, ethical reporting, and the credibility of its conclusions.

Reviewer #2: This study employs the TOE and TAM theoretical frameworks to explore the current application status, effectiveness, and challenges of AI technology through interviews with multiple e-commerce enterprises in Anhui Province. The research design is reasonable, the data is solid, and the findings offer certain insights. However, improvements are needed in theoretical depth, methodological rigor, and clarity of exposition.

1.Introduction Section:①The description of the research gap is somewhat broad and lacks systematic understanding. It does not precisely align with the design of this study.②The logical connection between the proposed multidimensional interaction of "technology-organization-environment-perception" and the integration of the TOE and TAM frameworks is unclear.

2.Literature Review:①It fails to review literature related to "stratified AI adoption," resulting in a lack of academic dialogue foundation for a core finding.②While it cites the TOE and TAM frameworks extensively, it lacks a critical discussion on how these two frameworks are combined at the organizational level. It is recommended to supplement the rationale for selecting these two frameworks, their complementarity, and their potential limitations in studying organizational AI adoption.

3.Methodology:①Sample information is incomplete. Details such as the industry type, revenue scale, and platform operation duration of the enterprises are missing.②The rationale for selecting Anhui Province is not provided. Can the level of e-commerce development in Anhui Province represent the general situation nationwide? A brief background explanation is needed to enhance the external validity of the sample.

4.Interview data findings & analysis:①A serious contradiction exists between Figure 2, where "Content Generation" accounts for only 6%, and the textual description in the results section stating that it is "widely and commonly used" by enterprises.

5.Discussion & Comparative Analysis:①The fifth key finding (dynamic adaptation mechanism) lacks a corresponding thematic section in the results, making its introduction in the discussion seem abrupt and poorly connected.②The comparison with literature lacks specific references. For instance, the claim that "existing literature assumes AI is a standardized solution" is not supported by citations to specific studies.

6.Conclusion:①The research limitations are not explicitly stated. It is recommended to supplement future research directions based on the gaps identified in this study.

7.Formatting and Details:Inconsistent formatting is present in the references section. Please strictly adhere to the journal's required style.

**Do you want your identity to be public for this peer review?** For information about this choice, including consent withdrawal, please see our Privacy Policy

Reviewer #1: No

Reviewer #2: No

---

## [Author Response · Author response to Decision Letter 1]

19 Jan 2026

Dear Editor and Reviewers,

Thank you very much for your time and effort in reviewing our manuscript titled " AI Adoption in E-commerce Enterprises: Insights into Current Practices and Future Directions from an Interview Study ". We truly appreciate the constructive comments and insightful suggestions provided by the reviewers, which have been immensely helpful in improving the quality and clarity of our paper.

We have carefully considered each of the points raised and have revised the manuscript accordingly. In the following sections, we provide a point-by-point response to the reviewers' comments. The changes in the revised manuscript have been highlighted using Red Color for your convenience.

Response to academic editor

1. The manuscript has been formatted in accordance with the requirements. However, the paragraph spacing has not been changed to double spacing for the sake of easier reading.

2. In this study, informed consent was obtained verbally from all participants prior to each interview. Before interview began, the interviewer explained the study purpose, voluntary nature of participation, confidentiality, and anonymization procedures using a standardized introductory script (see article Appendix). Participants were explicitly asked whether they agreed to participate and to be audio-recorded for academic research purposes. Participants’ agreement was captured on the audio recording at the beginning of each interview.

According to institutional guidelines, this study—based on professional interviews with enterprise representatives and involving no personal, medical, or vulnerable population data—was classified as minimal-risk social science research and did not require formal Institutional Review Board (IRB) approval. This exemption applies to studies collecting non-sensitive professional opinions without personal identifiers. We have clarified this procedure in the Methodology section accordingly.

3. We have revised the Funding Statement in accordance with PLOS ONE’s requirements. The amended Funding Statement is provided below for your update of the submission system:

Funding Statement:

This research was supported by the project “Application Research of Artificial Intelligence Technology in E-Commerce Enterprises in Anhui Province” (Fund Number: 2024AH052502). There was no additional external funding received for this study. The funder had no role in the study design, data collection and analysis, decision to publish, or preparation of the manuscript.

We confirm that no other institutional, commercial, or personal financial support was received during the conduct of this study.

4. We have removed all funding-related text from the Acknowledgments and all other sections of the manuscript, in accordance with PLOS ONE’s policy.

5. We confirm that our submission now includes the minimal data set required to replicate the findings of this qualitative study. We have deposited an anonymized dataset on Zenodo (DOI:10.5281/zenodo.18298170) containing a structured Excel file that summarizes all interview-derived evidence underlying the reported themes, figures, and conclusions. This dataset includes: The coded thematic summaries for each of the eight enterprises; The values used to construct all figures (such as, frequency of AI application domains, main platform); Metadata describing the operation and application status of the enterprise.

Full verbatim transcripts are not publicly shared due to ethical considerations and participant confidentiality. However, the deposited dataset represents the minimal de-identified data necessary to reproduce all analytical steps and results reported in the manuscript.

6. We have now finalized a data-sharing plan in accordance with PLOS ONE’s open data policy. All data underlying the findings have been deposited in a stable public repository (Zenodo) and are freely accessible at: (DOI:10.5281/zenodo.18298170). The dataset is fully anonymized and does not contain personal identifiers or sensitive information.

7. We have revised the manuscript to ensure that the ethics statement appears only in the Methodology section. All ethics-related text has been removed from any other sections of the manuscript.

Response to reviewer #1

1. We substantially strengthened the transparency of how interview data were transformed into themes and higher-level conclusions. First, we expanded the Methodology section to specify each step of the qualitative procedure, detailing how open codes were generated from raw transcripts, how they were aggregated into axial categories, and how these categories were mapped onto higher-order themes using TOE and TAM as sensitizing concepts. Second, in the Results section, we revised each thematic subsection to directly link major claims to their underlying codes and representative quotations from multiple enterprises. Third, we added summary tables that present enterprise profiles and the correspondence between codes, themes, and interview evidence. These tables provide a transparent basis for tracing how conclusions were grounded in the data.

2. In the revised manuscript, we substantially expanded the Methodology section to provide a transparent account of our qualitative procedures. We now specify that the thematic analysis followed a hybrid strategy, combining deductive and inductive coding. The TOE and TAM frameworks are explicitly described as sensitizing concepts guiding initial coding, while allowing new themes to emerge from the data. We further clarify how codes were developed, refined, and grouped into themes, and how these themes were linked to the analytical framework. To enhance trustworthiness, we detail the use of member checking, cross-review of coding by a second researcher. These additions enable readers to better evaluate the rigor, transparency, and robustness of the analytic process.

3. We have clarified the Data Availability Statement and now provide a de-identified dataset summarizing the interview materials as Supporting Information and in a public repository. This dataset constitutes the minimal data set required to reproduce all reported findings.

4. We have moved and expanded the ethics statement within the Methodology section. The revised text now explicitly describes the ethical pathway, including the use of verbal informed consent, how consent was documented (audio-recorded at the beginning of each interview), the anonymization procedures, data storage, and the basis for exemption from formal IRB review under institutional guidelines for professional, non-sensitive interviews. This ensures transparency and compliance with journal requirements.

5. We carefully edited the entire manuscript to improve clarity, precision, and readability in English. In addition, we added explicit statements in the ‘Sample Selection and Research Context’ section to clarify the limited sample size and the single-province setting, thereby tempering any unintended claims of broad representativeness. We also revised Figure 2 and its accompanying text to ensure that all quantitative-looking results are clearly presented as descriptive summaries of interview data.

Response to reviewer #2

1. Introduction Section

We have revised the Introduction to moderate the scope of the research gap and to ensure closer alignment with the study design. The revised version highlights the limited empirical insight into how AI applications are operationalized across platforms, processes, and enterprise contexts in e-commerce. The revised Introduction explains their complementary roles: TOE provides a structural perspective for examining technological, organizational, and environmental conditions, while TAM contributes a perceptual perspective for understanding how enterprises evaluate the effectiveness of AI applications in practice. This clarification supports the proposed multidimensional interaction of “technology–organization–environment–perception” and ensures a clearer alignment between the theoretical framework and the research design.

2. Literature Review

First, to respond to the absence of prior work on “stratified AI adoption,” we expanded Section ‘AI Adoption in E-commerce’ to incorporate multidisciplinary reviews, industry surveys, and SME-focused studies that document uneven and layered patterns of AI adoption across firms. The revised text now synthesizes evidence showing that AI diffusion and its benefits are distributed asymmetrically, shaped by firm size, digital readiness, and resource endowments.

Second, in response to the concern regarding the conceptual integration of TOE and TAM, we substantially revised Section ‘TOE & TAM in AI Adoption’. The revised section now explicitly clarifies:

(1) The rationale for selecting TOE and TAM—TOE captures structural and contextual conditions, while TAM explains evaluative perceptions shaping adoption;

(2) Their complementarity at the organizational level, where perceived usefulness and ease of use are interpreted as collective managerial evaluations rather than individual attitudes;

(3) Their analytical boundaries in organizational AI adoption research. We now explicitly acknowledge that TOE emphasizes contextual structures without explicating internal evaluation mechanisms, whereas TAM focuses on evaluative perceptions without modeling organizational resources or environmental constraints. The combined use of TOE and TAM is therefore positioned as an analytically balanced approach that links contextual conditions with organizational-level perceptions, while situating the analysis at the level of enterprise adoption decisions rather than broader inter-organizational dynamics or long-term capability evolution.

3. Methodology

We have revised the Methodology section to provide more complete and transparent sample information. A new table (Table 1) has been added to present the basic profiles of all interviewed enterprises, including business type, industry background, organizational scale, and key operational characteristics. This addition clarifies the heterogeneity of the sample and strengthens the basis for cross-case comparison.

In addition, we have expanded the description of the regional context to justify the selection of Anhui Province. The revised manuscript now explains that Anhui represents a transitional e-commerce environment with development indicators close to the national average, combining both emerging and mature digital practices. This positioning enhances the external validity of the study by situating the cases within a region that reflects common conditions faced by many Chinese e-commerce enterprises, rather than an extreme or atypical context.

4. Interview data findings & analysis

We have revised the figure by replacing the original pie chart with a bar chart to more accurately reflect the nature of the data. Because individual enterprises often apply AI across multiple functional areas, the values represent the number of enterprises that mentioned each application domain rather than proportional shares.

We have also added explicit textual clarification in Section ‘AI Applications in E-commerce Processes’ to explain the meaning of the figure and its statistical logic, emphasizing that the frequencies indicate the relative prevalence of AI use across domains.

5. Discussion & Comparative Analysis

(1) To address this issue, we have revised the structure of the Interview Data Findings & Analysis chapter by adding a new subsection ‘Application Pathways and Modes of AI Use’. This subsection is positioned before the analysis of application effectiveness and is explicitly grounded in interview evidence. It systematically reports how enterprises of different sizes integrate AI into their e-commerce operations Based on these findings, the fifth key point in the Discussion “The deployment of AI technology demonstrates a dynamic adaptation mechanism and a paradigm of human–machine collaboration” has been revised and now directly builds upon Section ‘Application Pathways and Modes of AI Use’. The discussion no longer introduces this concept in isolation but synthesizes patterns already identified in the Results. In addition, representative interview excerpts illustrating platform dependence, tool augmentation, and in-house development have been added to the interview summary table to strengthen the audit trail between data, themes, and conclusions.

(2) Based on your suggestions, we have added the missing references in Comparative Analysis: Interview Findings vs. Literature Review, and marked them.

6. Conclusion

We have revised the Conclusion section to explicitly articulate the limitations of the study, including the restricted sample scope, the qualitative and self-reported nature of the data, and the cross-sectional design. In addition, we have added a dedicated paragraph outlining concrete directions for future research, such as multi-regional expansion, mixed-methods validation, and longitudinal investigation of AI adoption.

7. Formatting and Details

By comparing with the formatting requirements, the references and other formats in the article were adjusted accordingly.

Once again, we would like to express our sincere gratitude to the reviewers for their invaluable feedback. We believe the manuscript has been significantly strengthened through this revision process. We hope that the revised version meets the high standards of PLOS ONE. Please do not hesitate to contact us if any further clarification or additional modification is required. We look forward to hearing from you.

Sincerely,

Tong Zhu, Mohd Zaidi Abd Rozan

---

## [Decision Letter · Decision Letter 1]

19 Feb 2026

Dear Dr. Zhu,

Thank you for submitting your manuscript to PLOS ONE. After careful consideration, we feel that it has merit but does not fully meet PLOS ONE’s publication criteria as it currently stands. Therefore, we invite you to submit a revised version of the manuscript that addresses the points raised during the review process.

I recommend that it should be revised taking into account the changes requested by the reviewers. Since the requested changes include valuable and constructive reviews, I would like to give you a chance to revise your manuscript. The revised manuscript will undergo the next round of review by same reviewers.

We look forward to receiving your revised manuscript.

Kind regards,

Baogui Xin, Ph.D.

Academic Editor

PLOS One

Journal Requirements:

Reviewers' comments:

Reviewer's Responses to Questions

**Comments to the Author**

Reviewer #1: All comments have been addressed

Reviewer #2: All comments have been addressed

2. Is the manuscript technically sound, and do the data support the conclusions?

Reviewer #1: Yes

Reviewer #2: Yes

3. Has the statistical analysis been performed appropriately and rigorously?

Reviewer #1: N/A

Reviewer #2: Yes

4. Have the authors made all data underlying the findings in their manuscript fully available?

Reviewer #1: Yes

Reviewer #2: Yes

5. Is the manuscript presented in an intelligible fashion and written in standard English?

Reviewer #1: Yes

Reviewer #2: Yes

Reviewer #1: (No Response)

Reviewer #2: (No Response)

**Do you want your identity to be public for this peer review?** For information about this choice, including consent withdrawal, please see our Privacy Policy

Reviewer #1: No

Reviewer #2: No

---

## [Author Response · Author response to Decision Letter 2]

21 Feb 2026

Dear Editor and Reviewers,

Thank you for your message and for the opportunity to submit the revised version of our manuscript. In this revision, we have carefully reviewed the manuscript in accordance with the journal’s requirements. Specifically:

1. We conducted a thorough check of the entire reference list to ensure completeness, formatting consistency, and accuracy.

2. We identified that one previously cited article (Reference 5) had been retracted. This reference has been removed and replaced with relevant reference that is recent in content.

3. We verified and supplemented DOI information where necessary to ensure that all references are properly documented.

We appreciate the constructive review process and thank you and the reviewers for your guidance. We look forward to your final decision.

Sincerely,

Tong Zhu, Mohd Zaidi Abd Rozan

---

## [Editor Report · Decision Letter 2]

23 Feb 2026

AI Adoption in E-commerce Enterprises: Insights into Current Practices and Future Directions from an Interview Study

PONE-D-25-56804R2

Dear Dr. Zhu,

We’re pleased to inform you that your manuscript has been judged scientifically suitable for publication and will be formally accepted for publication once it meets all outstanding technical requirements.

Kind regards,

Baogui Xin, Ph.D.

Academic Editor

PLOS One
---

## [Editor Report · Acceptance letter]

PONE-D-25-56804R2

PLOS One

Dear Dr. Zhu,

I'm pleased to inform you that your manuscript has been deemed suitable for publication in PLOS One. Congratulations! Your manuscript is now being handed over to our production team.

Kind regards,

on behalf of

Professor Baogui Xin

Academic Editor

PLOS One